# Places to Intervene in a Socio-Ecological System: A Blueprint for Transformational Change

**Teemu Koskimäki** 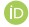

Crawford School of Public Policy, Australian National University, Canberra, ACT 2601, Australia; teemu.koskimaki@anu.edu.au

**Abstract:** The scientific community and many intergovernmental organizations are now calling for transformational change to the prevailing socioeconomic systems, to solve global environmental problems, and to achieve sustainable development. Leverage point frameworks that could facilitate such transformative system change have been created and are in use, but major issues remain. Scholars use the leverage point term in multiple contradicting ways, often confusing it with system outcomes or specific interventions. Accordingly, the underlying structural causes of unsustainability have received insufficient consideration in the proposed actions for transformational change. In this work, I address these issues by clarifying the definition for leverage points and by integrating them into a new blueprint for transformational change, with clarified structure and clearly defined transformational change terminology. I then theoretically demonstrate how the nine phases of the blueprint could be applied to both plan and implement transformational change in a socio-ecological system. Although the blueprint is designed to be applied for socio-ecological systems at national and international scales, it could also be applied to plan and implement transformational change in various sub-systems.

**Keywords:** structural change; SDG; post-growth; eco-anxiety; value change; root causes; solutions; backcasting

## 1. Introduction

Human actions throughout time, particularly after the industrial revolution, have placed ever growing amounts of pressure on environmental systems and as a consequence, some planetary boundaries have already been exceeded [1,2] and the Earth system is now on the precipice of exceeding several environmental tipping points [3,4]. Many of nature's supporting and regulatory functions have been compromised, which has led to global climate change [5] and the sixth mass extinction event in Earth's history [6]. Over the years, scientists have recognized and anticipated these ongoing global environmental problems and given repeated warnings calling for environmental action [7–11].

In response, nations have sought to address the environmental concerns concurrently with addressing issues related to human development, such as poverty and inequality, through the 2000–2015 Millennium Development Goals [12] and the successive 2030 Agenda for Sustainable Development [13]. Despite these efforts, the 2019 Global Sustainable Development Report (GSDR) on the progress towards the Sustainable Development Goals (SDGs) concluded that "the world is not on track for achieving most of the 169 targets that comprise the Goals" [14]. Climate change and biodiversity loss, together with rising inequalities and increasing waste outputs, were identified as issues "with cross-cutting impacts across the entire 2030 Agenda" that nonetheless "are not even moving in the right direction" [14].

In the latest global assessment report on biodiversity and ecosystem services, the Intergovernmental Science-Policy Platform on Biodiversity and Ecosystem Services (IPBES) argued that "current structures often inhibit sustainable development and actually represent the indirect drivers of biodiversity loss", which is why "fundamental, structural

change is called for" [15]. In this context, "structural" refers to the organization of system parts, processes, and rules, not to the sectoral composition of economies. This is also what the word "structural" refers to in the work at hand.

To solve the pressing global environmental problems and to achieve sustainable development, the scientific community and many intergovernmental organizations are now calling for "transformative" or "transformational" change (TC) [8,9,14–19], which the IPBES has defined as "A fundamental, system-wide reorganization across technological, economic and social factors, including paradigms, goals and values" [15].

The dictionary definition of the word "transform" includes three interpretations: (1) "to change in composition or structure"; (2) "to change the outward form or appearance of", and (3) "to change in character or condition" [20]. Therefore, TC can be understood as change that influences the system composition or structure, which then changes the character and condition of the system, which in turn changes its outcomes and outward appearance. This helps complement the IPBES definition, which lacked such causality. Hence, I define TC as fundamental and comprehensive structural change that influences the components and functions of a system, thereby changing its emergent outcomes. The components and functions that affect system outcomes include all technological, economic, and social factors (including values and goals).

Not only is it important to recognize the need for TC, but it is also necessary to understand how TC in socio-ecological systems could be achieved by utilizing specific points of leverage. Consequently, leverage points (LP) for transformational system change have been the focus of much research as of late, e.g., [21–25], and Chan et al. [24] have incorporated their version of the LPs in a framework for TC that is currently informing the IPBES [26,27] and the Convention on Biological Diversity (CBD) [18].

However, several major issues remain to be addressed. Scholars use the LP and other terms related to TC in multiple contradicting ways [22,23] and the underlying structural causes of unsustainability have received insufficient consideration, as too much emphasis is given to values and goals over structural realities.

In this theoretical work, I seek to address these issues and demonstrate how the LP approach could be improved, with literature-based and logical argumentation. In Section 2, I compare two influential LP frameworks, those of Meadows [28] and Chan et al. [24], to illustrate prevailing issues, to clarify the definition of LPs, and to demonstrate how the LPs could be applied to facilitate change in socio-ecological systems. In Section 3, I will integrate the LPs into an improved blueprint for TC, with clarified structure and TC terminology. I will then theoretically demonstrate how the nine phases of the TC blueprint could be applied to plan and implement TC in a socio-ecological system. In Section 4, I will discuss the findings of this work, giving particular focus to the role of values and goals in system change and the potential uses of the new blueprint. In the concluding Section 5, I emphasize the importance of clearly defining terms used in sustainability research and the importance of identifying and addressing the underlying causes of systemic problems to achieve global sustainability.

## 2. Leverage Points for Socio-Ecological System Change

An influential (and, to the best of my knowledge, the first) LP framework for system change was created by Donella Meadows in 1999, who defined LPs as "places within a complex system (a corporation, an economy, a living body, a city, an ecosystem) where a small shift in one thing can produce big changes in everything" [28]. With this framework, Meadows focused on all systems at an abstract level, identifying common shared properties that could be used to change the dynamics or outcomes a system and ranking these properties based on their potential power over the system.

Another influential development was made by Chan et al. in 2020 [24], who were inspired by the LPs defined by Meadows but considered them to be ill-suited for addressing complex global socio-ecological system change that has multiple contesting purposes [15,24]. Instead, through a process of iterative expert deliberation, the authors

identified eight LPs for societal transformation, which they defined as places "where to intervene to change social–ecological systems", and five levers, which they defined as "the means of realizing these changes, such as governance approaches and interventions", for implementing TC [24]. Table 1 presents a comparison of the items identified by Meadows and Chan et al.

**Table 1.** A comparison between the leverage points identified by Meadows [28] and the levers and leverage points identified by Chan et al. [24]. Leverage points: Priority points for intervention. Levers: Management interventions. The leverage points of Meadows are presented here in order of decreasing importance.

| Meadows (1999) | Chan et al. (2020) |
|---|---|
| **Leverage points** | **Leverage points** |
| 1. The power to transcend paradigms | 1. Visions of a good life |
| 2. The mindset or paradigm out of which the system—its goals, structure, rules, delays, parameters—arises | 2. Total consumption and waste |
| 3. The goals of the system | 3. Latent values of responsibility |
| 4. The power to add, change, evolve, or self-organize system structure | 4. Inequalities |
| 5. The rules of the system (such as incentives, punishments, constraints) | 5. Justice and inclusion in conservation |
| 6. The structure of information flows (who does and does not have access to what kinds of information) | 6. Externalities from trade and other telecouplings |
| 7. The gain around driving positive feedback loops | 7. Responsible technology, innovation, and investment |
| 8. The strength of negative feedback loops, relative to the impacts they are trying to correct against | 8. Education and knowledge generation and sharing |
| 9. The lengths of delays, relative to the rate of system change | **Levers** |
| 10. The structure of material stocks and flows (such as transport networks, population age structures) | A. Incentives and capacity building |
| 11. The sizes of buffers and other stabilizing stocks, relative to their flows | B. Coordination across sectors and jurisdictions |
| 12. Constants, parameters, numbers (such as subsidies, taxes, standards) | C. Pre-emptive action |
| | D. Adaptive decision-making |
| | E. Environmental law and implementation |

Being a part of the influential IPBES report [26], the framework of Chan et al. was also embraced by the CBD, whose Global Biodiversity Outlook 5 report stated that these levers and LPs "may be targeted by leaders in government, business, civil society and academia to spark transformative changes towards a more just and sustainable world" [18]. More recently still, the framework of Chan et al. was also used in the IPBES-IPCC co-sponsored workshop report on biodiversity and climate change [27].

Chan et al. referred to their work as a "framework of interventions" [24], and as such it has value. The authors detailed several areas where TC is needed, what actions could be taken to address specific problems, and they gave evidence-based guidance for decision makers on how these practices should be implemented [24].

My point of contest is with none of the above, but specifically with the way the LP term was used, and how the provided TC framework placed too much emphasis on values and gave too little consideration for other underlying structural causes of unsustainability in the socio-ecological system. The rest of this paper seeks to demonstrate why these issues should be addressed and how, starting with the LPs.

### 2.1. Clarifying the Definition of Leverage Points

Whereas the LPs of Meadows [28] were intentionally so broad that they could be applied to any social system at any scale, Chan et al. focused their LPs on the socio-ecological system primarily at the global to national scale [24]. However, unlike with the list of Meadows, most of the LPs listed by Chan et al. are not points of leverage that address the properties (components or processes) of the system, but instead they list some of the important outcomes of the system and areas where TC interventions are needed. For example, "total material consumption and waste" does not directly identify anything about the system itself that would need to change, or any specific actions. Instead, it implies that something needs to happen to reinforce and enable controlled changes to the levels of consumption and waste. Although Chan et al. accurately argue that total volumes of consumption and production must decrease among the wealthier countries and economic classes and increase among the more disadvantaged [24,29,30], they specified no actions or processes that could allow these changes to take place, beyond changes in values.

I have chosen to focus on the work of Chan et al. but they are by no means alone in misusing the LP term. In a recent special issue on "Leverage Points for Sustainability Transformations", Leventon et al. confirmed that authors use the term in multiple contradicting ways, writing: "It is evident through this collection that the papers do not always agree with how systems are (or should be) framed, nor use the same terminology to describe the fundamental components of the leverage points framework: the system, the lever, the leverage points, the interventions, etc. What is a leverage point for one author, is a system or an intervention for another" [22]. It is my view that this creates unnecessary confusion, which not only reduces the quality of the science, but that can also lead to adverse real-world consequences when the "LPs" are used to guide decision makers.

Meadows defined LPs as places "where a small shift in one thing can produce big changes" in the whole system [28]. Similarly, Chan et al. defined LPs as priority points for intervention, "where to intervene to change social–ecological systems" [24], and recently Linnér and Wibeck proposed to define LPs as "The part of the system that can be influenced for a proportionally greater effect on the whole system" [23]. However, these broad definitions may not be clear enough. When taken out of context, they can be interpreted as referring to anything small that can change something big. This is surely not what Meadows meant, which is why she defined the 12 specific LPs (Table 1). Authors should, therefore, always seek to refer to these 12 points, not just the overall definition. Furthermore, an intervention is not an LP. An intervention can tap into an LP when that intervention influences one or more of the specific system properties (LPs).

With the goal of clarifying this concept, I have rephrased the broad definition for LPs to the following form: LPs are key system properties where focused interventions can give rise to large changes in the behavior of a system. Here, the "key system properties" refer to the 12 points of Meadows [28].

A LP perspective can help understand how to create fundamental systems change towards sustainability [22], but only when that perspective correctly interprets what LPs are and what they are not, i.e., when the term is clearly defined and used. Defining terms clearly and using them consistently is what allows scientists to communicate effectively, within and outside academia. Using the LP term without defining it, or having multiple conflicting definitions for it, leads to problems. There is even a danger of turning a useful term into a buzzword devoid of any functional meaning.

### 2.2. Applying Leverage Points for Transformational Socio-Ecological Change

Chan et al. [24] decided to diverge from the LP typology of Meadows [28], because they deemed her typology to be ill suited to the context of complex global socio-ecological systems. However, in the following paragraphs, I show how the earlier typology of Meadows [28] not only can be used for this purpose, but also that applying it can reveal many valuable points and insights that were missing from the newer framework.

When comparing the framework of Chan et al. [24] to the LPs of Meadows [28], it seems that only two or three actual points of leverage are addressed by the newer framework of Chan et al. Firstly, "visions of a good life" can be seen to correspond to the paradigm shift in values that Meadows had high in her list (Table 1). Since societies have emerged from the interactions of minds, with each other and with the external world, and continue to be maintained by these minds, envisioning a new paradigm that facilitates the achievement of a good life in a new way can be a powerful factor in enabling socioeconomic changes that lead to justice and inclusion, changes in the levels of total material consumption and waste, reduced inequalities, and so forth. Secondly, "education and knowledge generation and sharing" is important in changing the minds of people and, thus, shifting shared societal goals towards achieving the new paradigm. These two LPs of Chan et al. address the first two LPs of Meadows. Third, the call to "unleash latent capabilities and relational values" may be interpreted as an inference to enabling a positive feedback loop that supports TC.

The highest of the unaddressed LPs was the goals of the system (LP 3 of Meadows). System goals are different from the shared societal goals, in that they emerge from the incentive structure of the system, not from the will (or values) of people—the two can conflict. To understand the structural incentives, it is important to identify the underlying structural causes that reinforce harmful or restrict beneficial behaviors within the system. Addressing such structural constraints is crucial for being able to utilize the other points of leverage covered by Meadows, such as applying critical changes to key parameters like taxes, subsidies, other policies, and the rules of the system. Key parameters are those that can influence the underlying structures and mechanisms, and critical means changes that surpass the normal range of variation for the parameter values, going beyond the status quo and leading to large changes in the whole system [28].

The points of Chan et al. (Table 1) can be seen as identifying some of the important areas such critical changes should seek to address. For example, "Externalities from trade and other telecouplings" can be addressed by implementing new negative feedback loops (LP 8 of Meadows), such as strong enough ("critical") cap-auction-trade systems for environmentally harmful inputs and outputs to help internalize externalities [31,32], and by adding new rules or changing existing parameters (LPs 5 and 12 of Meadows), such as ecological tariffs that influence trade [31,32]. Adding missing negative feedback loops helps balance the system into a new, more sustainable, state.

Restructuring or improving the information flows can also be relevant for transforming complex global socio-ecological systems (LP 6 of Meadows). Restructuring information flows means increasing information availability where it is relevant for the well-being of people and nature, to strengthen feedback, accountability, and public engagement. As Meadows identified, "Missing feedback is one of the most common causes of system malfunction" [28]. Due to it being relatively cheap and easy compared to other LPs, restructuring information flows should be combined with the other interventions early on in the transformation [28]. However, information alone is not enough unless it can affect influential feedback loops. For example, every year, societies are reminded earlier and earlier about the Earth Overshoot Day, but this has no visible impact on the system functions because the adaptive and correcting mechanisms of the prevailing system structure are too weak compared to the strong self-reinforcing feedbacks that maintain the prevailing harmful behavior.

The critical changes to key parameters and rules, implementation of new negative feedback loops, and information sharing can be supported by recognizing and targeting existing positive feedback loops in the system that act to reinforce old unsustainable patterns, which can create obstacles to change. One example of such is the positive "success to the successful" loop [28], which can lead to inequality and regulatory capture. With regulatory capture, agencies that should regulate the market become dominated by the industries they are supposed to regulate, with lobbying being one of the most visible manifestations of this process [15,19,33]. This creation and empowerment of vested interests [33,34] is just one way the system reinforces itself and creates resilience against change.

Even more common is the resistance of ordinary people to change, owing to how their livelihoods are often tied to the old unsustainable patterns, which creates a positive feedback loop where outdated structures support the short-term security and gain of people, who in turn wish to maintain the old structures. This discrepancy between long-term interests at the societal level and the short-term rewards at the individual level has long been recognized as a "social trap" [35–37]. In addition to material needs, the psychological needs and worldviews of people and businesses are also tied to the status quo, which has created a "social logic of consumerism" and materialism [38]. Not all positive feedback loops are harmful, however. They can also be strategically utilized, as Chan et al. [24] recognized, by creating new feedback loops that reinforce new sustainable patterns (LP 7 of Meadows).

Addressing stocks and flows, such as infrastructure and networks, is also relevant when seeking changes to the socio-ecological system (LP 10 of Meadows), as is ensuring

that size of stabilizing buffers, such as the extent and capacity of social security to abate the impacts of TC on employment levels and livelihoods (LP 11 of Meadows). Accounting for the length of delays relative to the rate of system change (LP 9 of Meadows) can also help prevent over- and understeering TC [28].

Changing the underlying structures (LP 4 of Meadows) with the help of higher order LPs is required to implement changes through the other LPs relating to system feedbacks, rules, stocks, buffers, and so on, and to align the goals of the system with the new societal goals. Such fundamental TC also provides an opportunity to add self-organizational capacity to the system that allows the system to adapt and evolve, increasing resilience and facilitating sustainable development in the long-term [28]. After a systemic and structural transformation, the policies that influence the levels of stocks, flows, constants, and (other) parameters can be optimized to increase socio-ecological-economic fairness and efficiency.

### 2.3. Summarizing and Focusing the Leverage Points

To provide a clear list of LPs that could be specifically used to guide transformational socio-ecological change in the context of complex international and national socio-ecological systems, I have reorganized and reduced Meadow's list of twelve LPs to the following five, based on the above discussion:

1. Societal goals—To lead and motivate transformation;
2. Structural goals—To address the underlying causes of problems;
3. Key parameters—To redirect the whole system with critical changes;
4. Information flows and feedback loops—To facilitate change and help overcome obstacles;
5. Flows, constants, and other parameters—To optimize a transformed system.

These are the places to intervene in a socio-ecological system, in order of decreasing importance. It must be emphasized that this ranking of LPs is based on their relative power over the system, not the sequence in which the points should be utilized. In fact, change makers may not have access to the higher LPs like structural goals right away, which is why they might have to start with information sharing and feedback loops first, thereby influencing key parameters and societal goals. After a sufficient demand (critical mass) is created, the structural goals can finally be addressed, which ultimately determine the system outcomes. As Fischer and Riechers [25] have emphasized, LPs can interact with each other, and sometimes deeper changes are needed for less powerful actions to work, whereas other times shallower changes can be used to pave the way for deeper changes.

### 3. A New Blueprint for Transformational Change

As a part of the IPBES report [26], Chan et al. [24] utilized their list of LPs to create an iterative framework for achieving TC (Figure 1). In it, they classified drivers of environmental problems and recognized various interventions and decision-making approaches that could be utilized to transform the socio-ecological system into one that is sustainable. The way Chan et al. visualized TC as an iterative process of interventions is valuable, as it demonstrates the dynamic and interlinked process of socio-ecological change (Figure 1). Importantly, Chan et al. also correctly identified that to achieve TC, focus should be expanded from direct drivers to indirect drivers [24].

This important and influential framework of Chan et al. [24] could be improved by adding in the missing LPs of Meadows and by clarifying the terminology in two ways: First, by making a clear separation between outcomes, specific interventions, and LPs, and second, by not calling the interventions "LPs", as argued in the previous section. Other shortcomings can also be identified: The framework lacks sufficient consideration for the underlying causes of global environmental problems, and it does not consider potential obstacles for TC. Lastly, the terms used in each step of the framework have not been clearly defined.

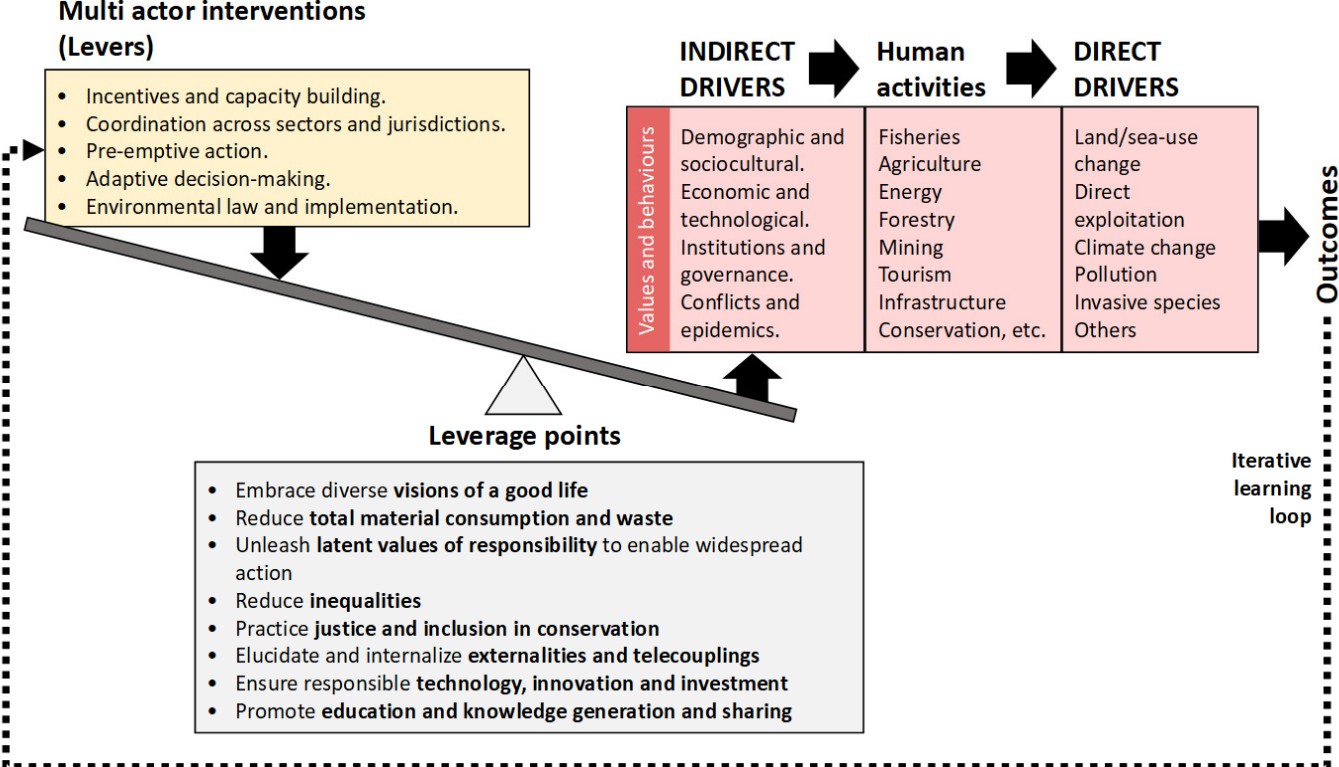

**Figure 1.** The transformational change framework of Chan et al. [24] and IPBES [26]. Everything else follows Chan et al. [24] except the box with drivers and human activities, as this part included more details in the earlier version [26]. Besides adding the word "Outcomes", which was signified with an illustration in the original figures, the text has not been altered in any way and has the original emphasis. Chan et al. [24] explained the emphasis in the leverage point box by writing that "At the leverage points (bolded), we have specified actions consistent with transformative change to sustainability (unbolded)". I have simplified the figure style to facilitate comparisons with Figure 2. Modified from [24,26].

In this section, I address the needed improvements by creating a new version of the framework of Chan et al. [24,26]. With this new "TC blueprint" (Figure 2), I add focus on the structural underlying causes of problems and the obstacles to change, beyond simply "values and behaviors". In addition to clarifying the structure and terminology, I also increase the applicability of the framework by dividing it into nine distinct and clearly defined phases, which are generalized enough to be applicable to any socio-ecological system in any context and at different scales.

The benefit of this new blueprint is that it clearly separates LPs from decision-making and actions, while also clearly categorizing and defining direct and indirect causes into threats, pressures, drivers, and the key underlying causes of problems. This schematic also helps visualize how "leveraging" change is not just a simple linear process. Instead, the process of this blueprint, with iterative feedback, helps account for irregularities, feedbacks, and other complex interactions of non-linear socio-ecological systems [23].

Comparing my blueprint (Figure 2) to the framework of Chan et al. (Figure 1), the general order in which stakeholders and decision makers use LPs to implement actions that influence indirect and direct causes of problems remains, as does the feedback between system outcomes and management (the "iterative learning loop"). However, this new blueprint is divided into nine specific phases, and the terminology is clarified in several respects compared to the earlier framework.

Firstly, I have included the list of five LPs summarizing Meadows's LP typology, and these are separated from interventions (phase 4) and outcomes (phase 1). I abstain from using the word "lever" alongside LPs in an effort to avoid unnecessary confusion arising from the similarity of the terms and their overlapping meanings in the English

language. Instead, the five levers of Chan et al. are included in phase 2, "decision-making and management".

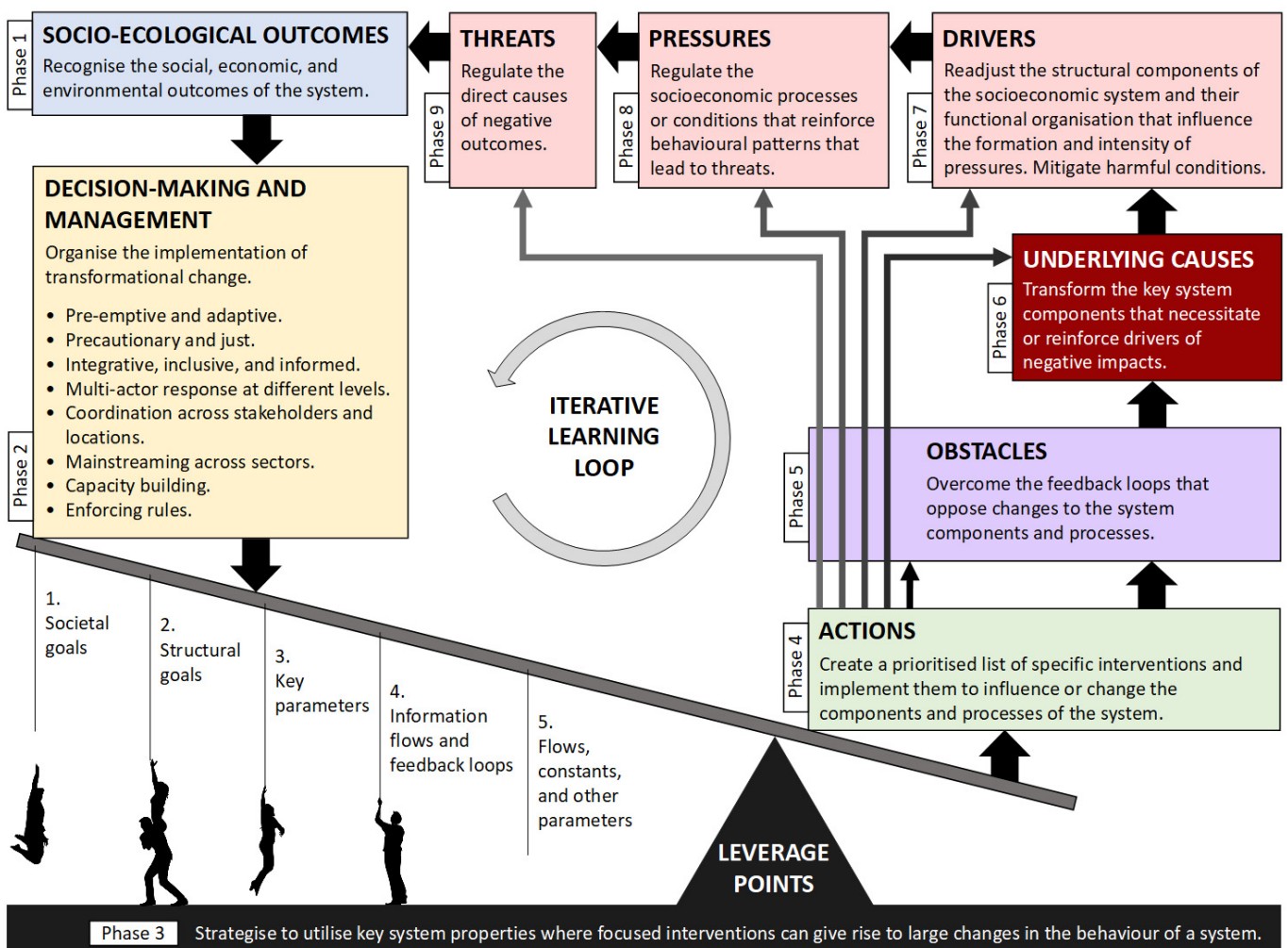

**Figure 2.** A new blueprint for transformational change. The nine phases guide the implementation (counterclockwise) or planning (clockwise) of directed and effective change to socio-ecological systems, using leverage points and addressing the underlying structural causes that can restrict system outcomes. Tables 2 and 3 provide definitions for the terms used in this blueprint.

Whereas Chan et al. used "multi-actor interventions" as another way to refer to their levers, in this new blueprint the word "intervention" refers only to actions, not decision-making and management (phase 2). Phase 2 is meant to organize the overall implementation of TC, whereas the actions (phase 4) specify the needed interventions that utilize the LPs of phase 3 to target phases 5–9.

When applying my blueprint in practice to a specific socio-ecological system, the needed interventions would be listed in the "Actions" phase, and the "LPs" of Chan et al. [24] could be used to identify and categorize important areas where actions are needed. Following the system dynamics terminology, in phase 3, I use the term "flows" to refer to the movement of matter, energy, or information through the system. "Constants" refer to variables that are set to some value and remain the same throughout time, whereas "parameters" are variables that can be altered or fine-tuned to guide the system behavior, such as subsidies, taxes, standards, and rules [28].

**Table 2.** Clarified terminology for transformational change.

| Phase | Term | Definition |
|:---:|:---:|:---:|
| 1 | Socio-ecological outcomes | Social, economic, and environmental outcomes of the system. |
| 2 | Decision-making and management | Organization of the implementation of transformational change in a way that seeks to ensure sustainable and desirable outcomes for the socio-ecological system. |
| 3 | Leverage points | Key system properties where focused interventions can give rise to large changes in the behavior of a system. |
| 4 | Actions | Specific interventions that influence or change the feedbacks, components, or processes of the system. |
| 5 | Obstacles | Feedback loops that oppose changes to the system components and processes. |
| 6 | Underlying causes | Key functions of specific structural components of the socioeconomic system that either cause drivers to lead to negative outcomes or that impede the operation of balancing feedback loops. I use the word "structural" to refer to the organization of system parts, processes, and rules, not to the sectoral composition of economies. |
| 7 | Drivers | The structural components of the socioeconomic system and their functional organization that influence the formation and intensity of pressures, and conditions (system dynamics) that restrict capacity for action. |
| 8 | Pressures | Socioeconomic processes or conditions that reinforce behavioral patterns that lead to direct threats. |
| 9 | Threats | Direct causes of negative outcomes. |

**Table 3.** Decision-making and management (phase 2) terms from the IPBES report [15,26], with new clarified definitions and reasoning.

| Term | Definition/Reasoning |
|:---:|:---:|
| Pre-emptive and adaptive | The potential outcomes of the planned changes are evaluated in advance and the actual outcomes are used to inform consequent actions. |
| Precautionary and just | Precautionary means that decisions are reviewed and made with caution, avoiding unnecessary risks, so as not to carelessly apply new innovations that may prove harmful in the long-term. Just means conforming to a standard of morality and correctness as defined by the group applying the blueprint (such as a nation). |
| Integrative, inclusive, and informed | Decision-making that is integrative and inclusive takes into consideration different perspectives and allows everyone to contribute to TC. Informed means decision-making considers the latest scientific knowledge of various disciplines and follows the best available multidisciplinary advice. |
| Multi-actor response at different levels | TC can be planned and applied by individuals, businesses, NGOs, governments, and other groups, each strengthening the overall societal effort to transform. |
| Coordination across stakeholders and locations | Response can be more effective when people and areas work together under shared overall goals. |
| Mainstreaming across sectors | Different governmental, societal, and economic subdivisions all integrate the same goals or practices of transformation into their agendas. |
| Capacity building | Allows more to be done while performing at a greater efficiency. A process that retains or improves the human, social, built, and natural capital that are needed to competently achieve needed changes. |
| Enforcing rules | The creation, modification, and implementation of laws, policies, and other guidelines, that are needed to change the behavior of systems and people. |

### 3.1. From Drivers to Underlying Causes

Importantly, with this new blueprint, I provide separate definitions for threats, pressures, drivers, and underlying causes (Table 2). In several previous studies, the word

"driver" has been used both in the context of direct and indirect influences [24,26,39]. While not strictly incorrect, it is better to clearly differentiate between the different levels of directness, because that can help reveal causalities and prioritize actions. In my blueprint, underlying causes are the most indirect, followed by drivers and then pressures, whereas threats are the only direct influences. The threats could also be referred to as direct drivers, as the prior studies have done, but this can create unnecessary confusion, which should be avoided as the blueprint is meant to guide not only academics, but also decision makers, who in turn may use it when communicating with the general public.

Previous research has identified several (indirect) drivers. For example, the IPBES [26] listed the following: demographic and sociocultural, economic and technological, institutions and governance, and conflicts and epidemics. The WWF [39] has similarly identified consumption, economics, institutions, governance, conflicts, technology, demographics, and epidemics as drivers, stating that "In the last 50 years our world has been transformed by an explosion in global trade, consumption and human population growth, as well as an enormous move towards urbanization. These underlying trends are driving the unrelenting destruction of nature". The formation and intensity of pressures, such as agricultural expansion and the use of non-renewable forms of energy, are influenced by these drivers.

The common characteristic of drivers is that they identify structural and functional properties of the socio-ecological system that can create harmful outcomes to people and nature, or more specifically, how the structural components of the socioeconomic system (e.g., people, businesses, governments) and their functional organization (interactions through institutions, patterns and levels of consumption, etc.) influence the formation and intensity of pressures. This includes conflicts and epidemics, although they could be seen as exceptional larger order system dynamics that only intermittently constrain the capacity of societies to work towards sustainability, albeit often severely.

So far, there has been little focus on the relative importance of the drivers, and even less on finding what structural components of the socioeconomic system cause drivers to lead to negative outcomes in the first place. For example, what causes consumption levels to exceed the limits of social and ecological sustainability? Why do governments subsidize harmful practices? Why is economic growth needed? Why does the global population keep growing? Why is technology used to exploit instead of regenerate? Instead of seeking structural answers for questions like these, studies tend to attribute some drivers, such as consumption, technology, or population growth, as the underlying causes, while explaining that the ultimate cause for them all is simply "values and behaviours" [15,24,26,39,40] (Figure 1). The confusing of LPs with system outcomes or specific interventions may have contributed to this insufficient consideration of the structural underlying causes in prior research.

By placing values and behaviors before drivers, the previous studies have inadvertently overlooked the importance of identifying what the key underlying structural causes are. Focusing on values and behaviors as the ultimate drivers of environmental problems can also have the effect of directing blame towards individual choice and away from structural realities [41], although problems such as overconsumption are driven by both outdated worldviews and structural requirements and reinforcements [38,42]. Considering the structural and underlying causes helps direct focus to the context in which behaviors occur.

By iteratively asking "why" the important indirect drivers exist and "what" causes them to lead to harmful outcomes, it is possible to identify and define a set of key underlying causes behind socio-ecological unsustainability. For example, the structural reliance or "societal addiction" [43,44] to economic growth has been identified as an underlying cause that not only maintains harmful behavioral reinforcements and restricts the available solution space, but also prevents the application of critical policy changes that would internalize externalities and correct telecouplings [31,38,40,44,45]. Similarly, the reliance of countries on international trade has been recognized as an underlying cause that drives down ecological and social standards, creates global inequality, and prevents countries from acting on sustainability [14,45]. It is due to the amplifying influence of these and other

specific underlying causes that drivers like governance failures, overconsumption, and population growth occur and lead to harmful outcomes. In my blueprint, underlying causes influence drivers, which create pressures, which in turn create direct threats (Figure 2). Furthermore, obstacles can exist that interfere with any action, regardless of what kind of "driver" the actions are directed to address (Figure 2). These are sometimes called "barriers" to change, but I have opted to use "obstacles" instead, which connotates more with something that can be overcome, even if it interferes with or slows down progress.

### 3.2. Decision-Making and Management Terminology

The items of phase 2 (decision-making and management) seek to aid decision makers and managers to organize the implementation of actions in a way that considers LPs and allows iterative TC to take place. All of these items directly correspond to those addressed and detailed in the IPBES report [15,26], although not all of them were included into the earlier figures (Figure 1). With my definitions (Table 3), and the new organization in the blueprint, I have merely sought to provide clarity to the points of this phase, to facilitate their implementation as best practice guidelines if the blueprint is used for planning and implementing TC.

### 3.3. Applying the New Blueprint

In theory, the nine phases of the blueprint could be applied to both plan and implement the transformational change of a socio-ecological system. Going through the nine phases of my blueprint, first the negative outcomes of the system are recognized in phase 1, which leads to a collective envisioning of new desired outcomes and an organization of a response in phase 2. Phase 2 is when the implementation of transformational change is planned and organized.

In this planning phase, the blueprint is applied clockwise to determine what directly threatens (phase 9) the desired outcomes, what pressures (phase 8) lead to the threats, what drivers (phase 7) cause the pressures, what ultimate and specific underlying causes (phase 6) cause the drivers to lead to pressures, and what specific actions (phase 4) would be needed to address the problems at each level, taking advantage of the LPs (phase 3). Then, potential obstacles (phase 5) are identified for each specified intervention, and actions are prioritized to address the obstacles first.

In this planning phase, the TC blueprint can be used as a part of a backcasting scenario building approach. In backcasting, criteria for a desirable future are defined first, after which a feasible and logical path is built from that future state to the present, which can help create alternatives otherwise not available through the forecasting of prevailing trends [46,47]. This makes backcasting particularly useful for considering how TC could be achieved [46].

When it comes time to implement the blueprint in practice, it is applied counterclockwise so that the system outcomes (phase 1) are used to justify and motivate the creation of new societal goals (phases 2 and 3), and the implementation of critical transformational actions (phase 4). First, the actions seek to overcome obstacles (phase 5) and fix the underlying root causes of problems (phase 6) that reinforce and necessitate behaviors that lead to the negative outcomes. Then, policies that directly address the drivers, pressures, and direct threats (phases 7–9) that impact people or nature in a negative way can be applied effectively and optimized. If the changes applied at each phase end up removing the threats to the desired social, ecological, and economic outcomes (phase 1), decision-making and management in phase 2 continues to enforce the new rules and maintain the new parameter values. If new threats emerge, the loop starts over and keeps going until the outcomes of the system are desirable.

This has to be the general order of action, because addressing the later phases (threats, pressures, and drivers) without first fixing the earlier phases (underlying causes and obstacles) is like swimming against a strong current [43]. Adding ad hoc fixes that work against the structural incentives creates inefficiency and wasted resources at best, and an

unsustainable fix at worst. So far, socio-ecological change has neither been sustainable nor transformational, because societies have neglected the most influential deep LPs [21,25] (1–4 in my summarized list) and actions have been directed to the threats, pressures, and drivers only, without addressing their underlying causes or properly accounting for the obstacles that create opposition to change.

## 4. Discussion

In this theoretical work, I provided clarification and improvements to existing LP frameworks that are currently informing international TC discourse and developed a new blueprint for TC. These contributions help provide more clarity on LPs and bring attention to the key underlying causes that cause drivers to lead to negative socio-ecological outcomes. Considering how the key underlying structural mechanisms behind unsustainable behaviors continue to be largely unaddressed in the sustainability discourse, not to mention in practice, it is unsurprising that efforts to achieve sustainability have so far not succeeded. The underlying causes need to be explicitly addressed and researched. Unless the TC discourse addresses this problem, it is unlikely to differ much from the sustainable development discourse in its success.

The key in creating transformational solutions that address the underlying causes is to first recognize, admit, and agree on the underlying causes, points of leverage, and the needed actions. Then, research efforts can be directed to identify potential obstacles for change and to model the likely outcomes of planned interventions. The nascent field of ecological macroeconomics has started to provide examples of relevant modelling work can test the social, ecological, and economic outcomes of TC policies, e.g., [48–51]. The actions have to be directed to overcome the strong self-reinforcing feedbacks that maintain the prevailing harmful behavior, and to implement new adaptive and correcting mechanisms to the system structure.

Given global impact inequalities among higher and lower income nations [19,45,52,53], it is particularly important to weaken the societal reliance on consumerism and growth in high-income nations while increasing the weight given to ecological and societal wellbeing considerations in decision-making at all levels [40,43,54–57]. In terms of sustainability, it is also worth to emphasize that for national and international decision-making to be "just", it must not further disadvantage the underprivileged or favor those already in positions of advantage, which would amplify harmful inequalities [11,15,26,58].

As a first step of the needed post-growth transformations in high-income nations, it is important to identify new measures for monitoring progress [55,59]. This can help societies evaluate if the system outcomes are within the "doughnut" in which social needs are met without exceeding planetary boundaries [60,61]. However, it must be emphasized that new indicators belong to phase 1 of the transformation and, although they are necessary, they alone are not sufficient for creating TC.

A shared characteristic in the frameworks of Meadows [28] and Chan et al. [24] was the emphasis on the importance of values. Chan et al. placed "Visions of a good life" and "Latent values of responsibility" high on the list of LPs. Similarly, second highest in Meadows's list was "The mindset or paradigm out of which the system—its goals, structure, rules, delays, parameters—arises". However, for the new values to have positive influence, they must be directed towards solving the underlying structural causes of problems.

Even if people understood that consumption and growth do not add to well-being beyond a certain point [59,62] and even if they embraced relational values and felt responsibility for taking care of the environment, that still would not be enough to implement TC, unless people feel a need to change the familiar but harmful socioeconomic structures. The new sustainable value systems can be viable in both appearance and practice only when they explicitly consider the underlying structural problems and how they could be solved without risking the security and well-being of citizens. Otherwise, people might not embrace the needed value shifts and solutions. Promoting a new ecologically and socially sustainable value system could even lead to an increase in paralyzing forms of eco-anxiety

or eco-anger [63], if the new value system does not direct people towards recognizing, demanding, and creating structural solutions to the underlying causes.

This important point can be clarified by modifying Meadows's bathtub analogy of a system. Consider the socio-ecological system as a bathtub that has too much hot water. Previously, when the water was considered too tepid, a system developed that effectively incentivized everyone to only run hot water. Consequently, a structural constraint was created to the faucet, which meant that later generations could not simply adjust the water to colder temperatures when the bath started to be too hot for comfort, even as the values and goals changed. The discord between observed reality and desired system state creates anguish and despair. Only when the structural problems with the faucet are fixed, can the lower temperatures (new goals) be achieved. Until then, attempts to change parameters (policies, taxation) can only determine whether the water is going to keep increasing in temperature rapidly or a bit slower, and new information and changing preferences can only keep increasing anxiety about the worsening situation, unless the emerging values are directed towards solving the structural problem.

Since all systems must adapt to TC, the blueprint offered here has a wide range of potential applications, and the fact that I have separated the LPs from specific outcomes or interventions improves the applicability of this blueprint compared to the earlier framework of Chan et al. [24]. One particularly important application for the blueprint would be to define what the causal hierarchy of global environmental problems is, focusing on establishing the underlying causes that countries would need to address. The TC blueprint could also be applied to solve problems in the social sphere, considering the specific threats and pressures that are reducing human well-being [13,60], listing the drivers that influence the formation and intensity of those threats and pressures, and identifying the key underlying causes that necessitate or reinforce the drivers, and then forming solutions that address those problems following the phases of the blueprint.

In addition, for socioeconomic systems to stay within environmental carrying capacity, the CBD [18] expects transitions in land-use, forestry, freshwater systems, fisheries and the use of oceans, agriculture and food systems, infrastructure, climate action, and health systems. The blueprint could be applied for planning and implementing TC in each of these sub-systems. Although solutions at the level of sub-systems cannot influence the underlying incentive structures of the socio-ecological system, the blueprint could be used to plan and implement changes that help improve the sub-systems and conform them to the larger TC occurring at the societal and international scales.

## 5. Conclusions

Since the late 20th century, we have been living in a "full world" where every additional unit of nature appropriated for human use presents trade-offs with dangerous consequences, which imposes new rules on socioeconomic systems [32,35,64]. To quote IPBES [34], "it is increasingly clear that structural, systemic change is necessary, and continuing along current trajectories increases the likelihood of disruptions, shocks and undesired systemic change".

In this work, I outlined how the LP frameworks for socio-ecological systems could be improved. The LP concept has been identified to be a "boundary object", which can provide an entry point for transdisciplinary and multi-stakeholder collaboration on complex system change [25]. To avoid creating further confusion, not only among scholars but also among the decision makers who the sustainability research aims to help, authors should always clearly define the terms they use. Specifically, they should clearly separate interventions and actions from LPs. Furthermore, reviewers of scientific manuscripts should make sure that if authors claim that some intervention addresses a LP, they must also argue how that intervention relates to the actual points of leverage originally recognized by Meadows, i.e., the key system properties where focused interventions can give rise to large changes in the behavior of a system. The LP term has specific meaning and should not be used as a buzzword.

After outlining the needed improvements to LP frameworks, I integrated them into a new blueprint for TC, with clarified terminology and structure. I then used the TC blueprint to theoretically demonstrate how its nine phases could be applied to plan and implement TC in a socio-ecological system. The blueprint is an improvement on previous frameworks, due to its clarified structure and terminology, and although it was designed for socio-ecological systems, it could also be applied to plan and implement TC in various sub-systems at different scales. I propose that the terminology I have clarified and defined in Tables 2 and 3 should become the new standard for TC discourse when addressing socio-ecological systems, which might make TC plans more approachable for a wider range of stakeholders.

Any set of solution proposals that seek to make the socioeconomic system ecologically and socially sustainable, seeking true TC, must systemically and successfully identify and address the underlying causes of the global problems. The blueprint I have developed could help academics and societies achieve this, helping to balance the social, ecological, and economic net-benefits of consumption, production, and trade, thereby bring the scale of economies into balance with Earth's carrying capacity. When combined with the policies and modelling tools developed in the field of ecological economics, this blueprint could help achieve the targets set for mitigating global environmental problems and for achieving sustainable development goals.

**Funding:** This research was funded by Kone Foundation, grant number 201804577.

**Acknowledgments:** I would like to thank my supervisors Ida Kubiszewski and Robert Costanza for their helpful feedback, and two anonymous reviewers for their valuable comments. I am particularly grateful for Henna E Virtanen, not only for her patient support and love, which keep me going, but also for her feedback which helped improve this work in many respects.

**Conflicts of Interest:** The authors declare no conflict of interest. The funders had no role in the design of the study; in the collection, analyses, or interpretation of data; in the writing of the manuscript, or in the decision to publish the results.

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
