# Peer review of "Places to Intervene in a Socio-Ecological System: A Blueprint for Transformational Change"

_sustainability, doi:10.3390/su13169474_

Round 1

Reviewer 1 Report

This might be an interesting and useful paper but there are aspects that create confusion.  For one, the introduction does not provide a clear 'road map' for readers as to what will be covered in the paper and why.  Second, I'm not clear how the blueprint was developed.  It could be appropriately done and well-grounded  - there might be a method to it - but I cannot discern this well.  Third, there are a variety of papers over the past several years that deal with this topic, and the paper could be better grounded in recent literature. Linner & Wibeck; Davelaar; Fischer & Riechers; etc.

Author Response

Below a point-by-point response to the specific issues raised by reviewer 1. Responses in blue.

This might be an interesting and useful paper but there are aspects that create confusion.  For one, the introduction does not provide a clear 'road map' for readers as to what will be covered in the paper and why. 

Thank you for pointing this out. I have improved the introduction as requested. Lines: 51-89.

Second, I'm not clear how the blueprint was developed. It could be appropriately done and well-grounded  - there might be a method to it - but I cannot discern this well. 

In lines 291-322, I now specify how I identified problems in the existing TC framework of Chan et al., and how I address the needed improvements by creating a new version of the framework, which is the TC blueprint. The paper is theoretical, so the method applied is literature-based and logical argumentation (I added a sentence to clearly state this in lines 76-77). This is how I created the list of 5 LPs, and ultimately how I developed the new blueprint.

Third, there are a variety of papers over the past several years that deal with this topic, and the paper could be better grounded in recent literature. Linner & Wibeck; Davelaar; Fischer & Riechers; etc.

Thank you for bringing my attention to these articles. I agree that it is important to give reference to recent developments in the field to help position my research in relation to it. To this end, I found the following articles to be relevant references for my paper and have now referenced them to further enhance the contextualization of my research: Fischer & Riechers (2019), Linnér & Wibeck (2021), and Birney (2021). I also added references to Keyßer & Lenzen (2021) and Hickel & Kallis (2020).

Reviewer 2 Report

A very interesting article from a theoretical point of view. However, in my opinion, this is its drawback. The article could have looked more advantageous if it had been diluted with practical examples of all the author's reasoning and generalizations. Maybe it resonated with a wider audience.

Author Response

Below a point-by-point response to the specific issues raised by reviewer 2. Response in blue.

A very interesting article from a theoretical point of view. However, in my opinion, this is its drawback. The article could have looked more advantageous if it had been diluted with practical examples of all the author's reasoning and generalizations. Maybe it resonated with a wider audience.

Thank you for the compliment and feedback. The specific purpose of the current paper is to show how existing frameworks can be improved, and to provide the theoretical background for the blueprint. The main audience is sustainability scholars. I agree that practical examples would be beneficial for demonstrating and testing the blueprint, and that is going to be the focus of future research. The current paper is purposefully focused in-depth on the theory. I improved the introduction, clarity of methods and results, and added references to improve the contextualization of my research. In particular, see lines: 51-89, 291-322, and 76-77.

Round 2

Reviewer 1 Report

Manuscript is much improved.  Thank you for your close attention to the recommendations. 

Reviewer 2 Report

I wish you success in your further research on this topic.